# Mapping the Solastalgia Literature: A Scoping Review Study

**DOI:** 10.3390/ijerph16152662

**Published:** 2019-07-25

**Authors:** Lindsay P. Galway, Thomas Beery, Kelsey Jones-Casey, Kirsti Tasala

**Affiliations:** 1Department of Health Sciences, Lakehead University, Thunder Bay, ON P7B 5E1, Canada; 2Faculty for Natural Sciences, Kristianstad University, 291 88 Kristianstad, Sweden; 3Faculty for Teacher Training, Kristianstad University, 291 88 Kristianstad, Sweden; 4Weave Collaborative, Duluth, MN 55805, USA

**Keywords:** solastalgia, mental health, emotional health, place, climate change, environmental change, landscape

## Abstract

Solastalgia is a relatively new concept for understanding the links between human and ecosystem health, specifically, the cumulative impacts of climatic and environmental change on mental, emotional, and spiritual health. Given the speed and scale of climate change alongside biodiversity loss, pollution, deforestation, unbridled resource extraction, and other environmental challenges, more and more people will experience solastalgia. This study reviewed 15 years of scholarly literature on solastalgia using a scoping review process. Our goal was to advance conceptual clarity, synthesize the literature, and identify priorities for future research. Four specific questions guided the review process: (1) How is solastalgia conceptualized and applied in the literature?; (2) How is solastalgia experienced and measured in the literature?; (3) How is ‘place’ understood in the solastalgia literature?; and (4) Does the current body of literature on solastalgia engage with Indigenous worldviews and experiences? Overall, we find there is a need for additional research employing diverse methodologies, across a greater diversity of people and places, and conducted in collaboration with affected populations and potential knowledge, alongside greater attention to the practical implications and applications of solastalgia research. We also call for continued efforts to advance conceptual clarity and theoretical foundations. Key outcomes of this study include our use of the landscape construct in relation to solastalgia and a call to better understand Indigenous peoples’ lived experiences of landscape transformation and degradation in the context of historical traumas.

## 1. Introduction

Solastalgia, preliminarily and broadly defined as the distress caused by the transformation and degradation of one’s home environment [1,2,3], is a relatively new concept with particular relevance to the environment–health–place nexus. Given the current speed and scale of climatic and environmental change, and the increasingly urgent calls to better understand the links between changing environments and human health, solastalgia merits further investigation and theoretical development [4,5,6]. As Louv [7] argues, “if climate change occurs at the rate that some scientists believe it will, and if human beings continue to crowd into de-natured cities, then solastalgia will contribute to a quickening spiral of mental illness” [7].

Now is a critical time for an in-depth consideration of solastalgia. Individuals and communities worldwide are increasingly witnessing the dramatic and chronic degradation of environments and experiencing a range of associated responses and impacts. Although the physical health implications of climatic and environmental change are increasingly well documented [8,9,10], the emotional, mental, and spiritual health implications remain understudied [4,10,11,12,13]. Climate change is no longer a future threat, and we are witnessing a convergence of attention, an international wake-up call, to the perils of climatic and environmental change [14]. For example, consider the recent nexus of field reports of climate change, international climate reporting, and global protests. In October 2018, the Intergovernmental Panel on Climate Change issued a special report on the impacts of global warming of 1.5 °C [14]. Soon after, the Fourth National Climate Assessment was issued by 13 U.S. federal agencies and presented a stark warning for climate change consequences [15]. In December 2018, the 24th Conference of the Parties to the United Nations Framework Convention on Climate Change (COP24) outcomes highlighted the importance of staying within the 1.5 °C temperature rise target, as well as planning for how to achieve this goal [16]. At the core of this mounting evidence, the growing media attention, and the increasingly common lived experiences of climatic and environmental change, comes growing distress and a range of mental, emotional, and spiritual health consequences. 

The overarching goal of this study is to characterize and synthesize the scholarly literature on solastalgia using a scoping review process. Scoping reviews are a form of knowledge synthesis used to map existing literature on a specific topic and are particularly useful when a topic is complex and heterogeneous [17,18]. Scoping reviews aim to examine a body of literature on a given topic by contextualizing knowledge, mapping the key concepts underpinning a research area, and identifying the primary sources and types of evidence available [19]. In doing so, scoping reviews advance conceptual clarity and identify gaps in the literature, offering valuable insight for future research directions [20].

In this scoping review, we focus on the scholarly literature that uses solastalgia as a lens for examining emotional, mental, and spiritual health dimensions of climatic and environmental change. Although there is a range of related concepts used to understand environmentally-induced distress and the associated health impacts, ecological grief and eco-anxiety as key examples, we have not reviewed any such related concepts here. A key theoretical aspect of solastalgia that sets it apart from related concepts is an explicit focus on place: solastalgia is a place-based lived experience. In the paper that follows, we describe the scoping review process and findings and subsequently discuss the implications of these findings. We also make recommendations for advancing future solastalgia-related research. 

## 2. Materials and Methods 

A scoping review, one of many different forms of knowledge synthesis, was identified as the most appropriate review method for mapping the solastalgia literature for several reasons. First, solastalgia is a relatively new concept, and scoping reviews are most useful for examining literature that has not been previously reviewed [17]. Second, scoping reviews are useful for bodies of research that cross disciplinary and methodological boundaries, which is undoubtedly the case with the solastalgia literature. Third, scoping reviews are helpful for identifying research gaps and emerging research priorities. This review was guided by the framework for conducting scoping reviews outlined by Arksey and O’Malley [20], while also considering suggestions from Levac et al. [21]. Therefore, we applied a six-phase process: (1) identifying the research question(s); (2) identifying relevant studies; (3) selecting studies for inclusion in the review; (4) data charting; (5) collating, synthesizing, summarizing, and reporting results; and (6) consulting with other scholars, practitioners, and potential knowledge-users [20]. It is important to note that in practice, these phases unfold iteratively rather than linearly. Also noteworthy, following the recommendation by Levac et al. [21], the review team met regularly to review progress, address challenges, and discuss discrepancies. 

Four specific research questions guided the scoping review process: 

(1) How is solastalgia conceptualized and applied in the literature? 

(2) How is solastalgia experienced and measured in the literature? 

(3) How is ‘place’ understood in the solastalgia literature? 

(4) Does the current body of literature on solastalgia engage with Indigenous worldviews and experiences? 

### 2.1. Data Sources and Search Strategy

With the help of an experienced health sciences librarian, our team systematically searched 10 electronic databases using the keyword ‘solastalgia’ (in title, keywords, subject, and abstract). Specifically, we explored the following electronic databases: Web of Science, Pubmed, PsycINFO, GreenFile, PhilPapers, ProQuest, Scholar Portal Books, Embase, Academic Search Premier, and Academic OneFile. This set of databases was identified to cast a wide net and to ensure identification of all papers in the published scholarly literature concerning the concept of solastalgia in our search. Grey literature was not included in our search process. The database search was conducted in October 2018. The database search identified a total of 95 papers; after removing 46 duplicates, 49 papers remained. An adapted version of the Preferred Reporting Items for Systematic Reviews and Meta-Analyses (PRISMA) flow diagram is presented in Figure 1 in the Results section. 

### 2.2. Relevance Screening and Eligibility 

The remaining 49 articles were screened for relevance by reading titles and abstracts; a process which resulted in the exclusion of 3 additional citations deemed as irrelevant. Furthermore, the reference lists of the 46 remaining papers were manually searched to identify any additional citations missed from the database search. An additional 5 citations were added following the full reference list review for a total of 51 papers. Subsequently, eligibility was assessed by two independent reviewers using as a priori developed inclusion and exclusion criteria. The inclusion and exclusion criteria outlined in Table 1 were applied to the set of 51 papers by reading abstracts and the full text. Discussion was used to resolve any difference of opinion between the two independent reviewers. Inter-rater agreement was assessed using percentage agreement between the two reviewers. Percentage agreement for article eligibility was 94.11%; more than 90% percentage agreement is generally considered sufficiently high to continue with a review process. A total of 29 articles met the inclusion and exclusion criteria and were included in the data charting and synthesis process.

### 2.3. Data Charting and Synthesis

The four research questions guided the data charting and synthesis process, and a data extraction form that was co-created by all review team members. Two reviewers independently pilot-tested the data extraction form using a set of four sample articles. The two reviewers then assessed the data extraction form for completeness, clarity, and ease of use. The form was modified based on the pilot-testing process. The final data extraction form included five main categories: (1) general characteristics of the paper; (2) conceptualization and application of solastalgia; (3) experience and measurement of solastalgia; (4) how place is understood; and (5) engagement with Indigenous worldviews and experiences. Each of these main categories contained multiple specific items for coding using either verbatim text extraction, textual description, categorical coding, or binary coding. (see Table A1 for the full data extraction form).

The full review team extracted data for each article using the finalized data extraction form and Google sheets software. For each paper in our final sample, one member of the review team acted as ‘primary reviewer’ and a second member of the review team acted as the ‘verifier.’ The primary reviewer and the verifier discussed discrepancies in their respective data charting processes and resolved any discrepancies through discussion. Finally, the reviewers used extracted data to tabulate, summarize, and synthesize the content using frequencies, percentages, tables, and textual analyses where appropriate. 

### 2.4. Consultation Process 

Consultation with other scholars, practitioners, and potential knowledge-users (both settler and indigenous) was an important part of this study. The development of our specific research questions was informed in large part by knowledge gaps and priorities identified at a workshop entitled “Solastalgia: Investigating the nexus of climate change, place, and human well-being” which took place in October 2018 in Minnesota, USA. This workshop was inspired by place-based climate change research in the Western Lake Superior region [22,23,24] including the North Shore Climate readiness project and the Twin Ports Climate Conversations [25] and was supported by funding from the University of Minnesota’s “On the Horizon” program. Participants included a diverse group of researchers and practitioners working on issues at the environment–health–place nexus in the Western Lake Superior region, and interested in exploring the concept of solastalgia, as well as research and practice concerning solastalgia more specifically. 

From the first workshop in October 2018, a more formal group formed, now calling itself the International Solastalgia Collaborative. “International” to acknowledge participation by First Nations people, Canadians, and Americans (and institutions from those nations). The International Solastalgia Collaborative met for a second workshop in January 2019 in Superior, Wisconsin. During the workshop, the authors of this paper presented preliminary findings of the scoping review and received criticism and feedback in a large group discussion. Feedback collected during the consultation process was documented and incorporated into the discussion and recommendations for further research presented in this paper. Final consultation took place in June 2019. The group was also invited to review and make comments on the manuscript at several stages throughout its development.

## 3. Results

From the 49 unique citations identified in the initial database search; the screening and eligibility assessments resulted in a final set of 29 (See Figure 1 and Appendix A (Table A1 and Table A2) for complete list). In the following sections, we summarize the solastalgia literature in terms of general characteristics and concerning our set of four research questions. 

### 3.1. General Characteristics of Papers

Key findings from exploration of the general characteristics of the solastalgia literature show the literature to be recent, multidisciplinary, and empirical in nature, and further, the Australian origins of the work is clear. All papers were published between 2004 and 2018; our results indicate that the number of published scholarly papers has increased over these 15 years (See Figure 2). The concept of solastalgia was first introduced by Dr. Glenn Albrecht in 2003 at an Ecohealth Forum in Montreal, Canada, so it is not surprising that 2004 is the year when we begin to see publications. The year 2018 had the largest number of papers published (17%) suggesting a growing interest in solastalgia since the early 2000s. It should be noted that our review only covered articles published between January and October 2018, and therefore did not capture the full year. The use of the concept is not confined to a specific discipline; the solastalgia literature spans a wide range of academic disciplines, including public health, human geography, anthropology, and philosophy. Concerning geographic distribution among the papers in our final sample, there is a focus on Australia, which comprised half of all articles while another 19% of articles focused on the United States. The large proportion of solastalgia-related papers focused on Australia may be explained by the fact that more than half of the lead authors are situated in Australia and because Albrecht, who coined the term and still leads much of the work and discourse around solastalgia, resides in Australia. Countries in the Global South were underrepresented.

We coded papers in our review as ‘empirical,’ ‘theoretical/conceptual,’ ‘review papers,’ or ‘practice/policy focused papers.’ The majority were empirical papers (59%), 24% were theoretical or conceptual papers, and 17% were review papers. None were coded as practice/policy focused papers.

### 3.2. How is Solastalgia Conceptualized and Applied in the Literature?

Through the scoping review process, we sought to advance our understanding of how solastalgia is conceptualized and applied in the published literature. The data charting form included eight specific items to assist in answering the question (see Table A1). We examined how authors define solastalgia in their work using textual analysis of verbatim definitions employed. Common elements of definition of solastalgia included: (i) a description of the transformation of the environment (i.e., unwelcome environmental change associated with resource extraction); (ii) a description of the place/environment being transformed (i.e., one’s valued home environment); and (iii) mention of specific impact(s) associated with the transformation (i.e., mental health impacts, depression). Textual analysis of the verbatim definitions illustrates that the words ‘environment’, ‘home,’ ‘distress,’ and ‘loss’ are the most commonly employed words when defining and describing solastalgia. Albrecht and his influential paper titled “‘Solastalgia’: A New Concept in Health and Identity” [1] is commonly referenced by the papers in our review when defining solastalgia. 

We also sought to gain a more in-depth understanding of the sources of environmental change resulting in solastalgic experiences as well as the terms employed to describe these sources. We found that there is a diversity of factors triggering solastalgia including extreme weather events, (e.g., Warsini [26]), resource extraction (e.g., Canu [27]), climate change (e.g., Tschekart et al. [28]), and political violence (e.g., Sousa [29]) (See Table 2). Extreme weather events such as flooding, drought, or earthquakes are the most commonly considered sources of environmental change described in the literature. Some of the factors causing the negative transformation of places ‘acute’ in nature, leading to dramatic and sudden transformation with ongoing impacts, while others are ‘chronic’ and result in more gradual and subtle degradation of places. Climate change is also a commonly examined cause of environmental degradation and transformation; interestingly, climate change leads to both acute impacts (e.g., floods and wildfires) and chronic degradation of places (e.g., sea level rise). Examples of terms commonly used to describe the environmental change include: ‘cumulative’, ‘compounding’, ‘imposed’, ‘unwelcome’, ‘unwanted’, ‘dramatic’, ‘negative’, and ‘profound/intense’ [1,30,31,32,33,34,35,36,37,38].

In terms of understanding how solastalgia is conceptualized and applied, we also examined whether the papers in our review describe related concepts—i.e., alternative concepts used to understand distress and associated emotional, spiritual, or mental health implications of changing environments—and if so, which concepts specifically. Solastalgia is part of an emerging set of concepts aimed at describing and accounting for the implications of climatic and environmental change on mental health and wellness. This emerging set of related emotional concepts has been described as “psychoterratic” or earth-related states [30,39]. Related concepts most commonly described in the solastalgia literature include ecological grief i.e., “the grief felt in relation to experienced or anticipated ecological losses including the loss of species, ecosystems, and meaningful landscapes due to acute or chronic environmental change” [33], and eco-anxiety, i.e., “anxiety related to a changing and uncertain environment...people become overwhelmed by the sheer scale, complexity and ‘wickedness’ of the problems we are facing" [31]. Other concepts discussed, but less commonly, include topophilia [40], nature-deficit disorder [41], and eco-paralysis. Most (75%) of the papers we reviewed make mention of at least one additional related concept. However, few papers provide clear definitions of these related concepts or outline explicitly the similarities and differences between solastalgia and these associated concepts leading to a lack of conceptual clarity in the literature overall. 

### 3.3. How is Solastalgia Experienced and Measured in the Literature?

A series of items were included in this scoping review to consider the details of solastalgic experiences, who is experiencing solastalgia, and how it is measured. To better understand the measurement of solastalgia, we focused on the subset of empirical studies, specifically (n = 17). Study populations were quite varied and ranged from geographically based residents of specific cities or regions, culturally identified subpopulations such as Palestinian women of the West Bank, and victims of natural disasters, such as volcano survivors, etc. Overall, there is a focus on individuals and groups that depend on and/or live close to the land, farmers for example. Notably, a single empirical study focused on youth as the specific study population of interest.

We found that a variety of approaches were used to measure solastalgic experiences in the sub-set of empirical studies. The most common was the use of qualitative interviews (n = 12). Five studies used surveys or scales. The Environmental Distress Scale (EDS), developed and validated by Higginbotham et al. [3], is a key example. The EDS was informed by qualitative fieldwork in Australia’s Upper Hunter Valley, a region where open-cut coal mining has dramatically altered and degraded landscapes [3]. One of the six components of the EDS specifically measures feelings of solastalgia using a set of nine Likert items (see Table 3) [3]; these items were used by the researchers to map the dimensions of solastalgia. The EDS was applied in two other empirical studies examining the impacts of living in an environment degraded by an extreme weather event [34,42]. A subscale of the EDS was also used in one additional study [43]. 

With regards to the lived experiences of solastalgia as described in the literature, we examined whether gender dimensions were explicitly considered (using a binary yes/no item). Results indicate that only three papers investigated gender as it pertains to solastalgia in a meaningful way. One example of a paper with an explicit gendered focus is McNamara and Westoby [44] who considered responses to climate change among Elder women in a Pacific Island community in their study titled “Solastalgia and the Gendered Nature of Climate Change: An Example from Erub Island, Torres Strait”. We also sought to gain an understanding of the mediating factors theorized and measured in the literature (i.e., factors that affect the presence or degree of solastalgia in response to climatic or environmental change). Common mediating factors identified in our review include: sense of place, place attachment, powerlessness/sense of power, trust in government or industry, and uncertainty about the future. The focus on these mediating factors is likely informed by the conceptual framework outlined in Higginbotham et al.’s [3] foundational paper which described environmental distress and human distress as a cycle where sense of place trust in government and trust industry were hypothesized as key mediating factors in response to environmentally-induced distress. Specific measurement of mediating factors, and quantitative work using population-based studies in particular, is lacking from the literature at present.

### 3.4. How is ‘Place’ Understood in the Solastalgia Literature?

Nearly all papers in this review (27/29) specifically describe solastalgia as a place-based phenomenon. Interestingly, despite the centrality of place in relation to solastalgia, few authors clearly and explicitly define what they mean by ‘place’. We found that only 14% of the reviewed articles provided a specific definition of place as applied in their work. We also found that only five papers clearly articulate a specific place theory. Moreover, ambiguity across key people–place concepts (i.e., sense of place, place attachment, and connectedness to place) is common.

Authors conceptualized ‘place’ in three main ways: (1) as a social-ecological concept; (2) as an ecological or environmental concept; or (3) as a political or geographic concept. The majority (66%) of the papers reviewed conceptualized place as a social-ecological construct. This idea is generally defined as an integrated perspective of humans-in-nature and part of place [45]. For example, Tschakert et al. [28] considered embodied experiences of environmental and climatic changes in landscapes of everyday life in Ghana and clearly articulated the interconnectedness between ecological systems and social systems. In nearly one-quarter of papers in this review, place was conceptualized as an ecological or environmental construct, i.e., as “a neutral backdrop or setting for human activity” [46]. The remaining 14% conceptualized place using a geographic boundary or political definition, defining place primarily by geography and location. We also found that place is described in myriad ways such as “ecosystem” or “traditional land” [30]. ‘Landscape’ is another term used by authors to describe place. For example, Canter describe landscape as the “interplay among the physical attributes of an area, people’s conceptions and interpretations, and their actions and activities within the physical setting’’ [47]. Similarly, the landscape idea was captured by Pannell via the use of the term “geo-cultural landscape” [38]. The use of landscape emphasizes the interrelatedness between biophysical settings and social, cultural, or interpersonal meaning [2] while also articulating that places are imbued with meaning [48]. A specific aspect of meaning that is evident in the literature is the idea of belonging and attachment. Also noted was how people–place relationships may affect susceptibility; may make one more susceptible to distress from environmental or climatic change in other words [33].

### 3.5. How does the Literature on Solastalgia Engage with Indigenous Worldviews and Experiences?

The review team used binary codes to categorize papers according to whether or not they (i) include any discussion or analysis about how solastalgia relates to Indigenous worldview; (ii) substantively consider Indigenous people/communities in discussions of solastalgia, and (iii) discuss and/or account for the unique lived experiences of Indigenous peoples/communities. After the first round of coding and preliminary analysis, and after receiving feedback from the members of the International Solastalgia Collaborative, an additional item was added: a binary code to identify whether the papers’ research was conducted by Indigenous peoples and/or in partnership with Indigenous peoples (coded for empirical papers specifically).

Of the 29 papers reviewed, nearly all (approximately 90%) examined solastalgia either through a western worldview or through multiple worldviews, rather than using an Indigenous worldview specifically (or multiple Indigenous worldviews). While none of the papers explicitly defined Indigenous versus western worldviews, our review team understood Indigenous worldview(s) to mean how Indigenous people understand the way the world works, and their relationship to it. Despite the diversity across Indigenous populations worldwide, many Indigenous scholars have attempted to articulate the common features across worldviews shared by Indigenous peoples across, including Simpson [49], who outlined seven principles of Indigenous worldviews in her 2000 paper Anishinaabe ways of knowing. Knowledge is holistic, cyclic, and dependent upon relationships and connections to living and non-living beings and entities. Second, there are many truths, and these truths are dependent upon individual experiences. Third, everything is alive. Fourth, all things are equal. Fifth, the land is sacred. Sixth, the relationship between people and the spiritual world is important. Seventh, human beings are the least important in the world [50].

Almost one-fourth (24%) of the papers considered the unique lived experiences of Indigenous people and communities when discussing or describing solastalgia. Most of these discussions were rather general descriptions of Indigenous cultures and their relationships to land. For example, McManus et al. [51] explain that “the positive love of landscape and place is typically a more intense feeling/emotion for Indigenous people and people such as rural and remote folk who live closely to the land/soil. When a much-loved landscape is desolated, an equally powerful negative feeling/emotion is likely to be experienced” [51]. Throughout this paper we use the term “Indigenous” to refer to “communities, peoples and nations which, having a historical continuity with pre-invasion and pre-colonial societies that developed on their territories, consider themselves distinct from other sectors of the societies now prevailing on those territories, or parts of them. They form at present non-dominant sectors of society and are determined to preserve, develop and transmit to future generations their ancestral territories, and their ethnic identity, as the basis of their continued existence as peoples, in accordance with their own cultural patterns, social institutions and legal system” [52]. However, it is important to note that, “Indigenous peoples have argued against the adoption of a formal definition at the international level, stressing the need for flexibility and for respecting the desire and the right of each Indigenous people to define themselves,” and therefore we have not imposed any definitions of indigeneity here [53].

However, several papers do describe specific experiences of solastalgia by Indigenous peoples. McNamara and Westoby [44], for example, focus on the experiences of respected elders—women called ‘aunties’—on Erub Island of Torres Strait, concerning climate change specifically. In addition to outlining specific emotions, feelings, and mental health implications experienced by these women, the authors also assert that, “an intrinsic synergistic relationship connects the health of Islanders and the well-being of their land and sea country. Consequently, biophysical impacts have the potential to affect mental health in ways not often considered in non-Indigenous societies” [44]. 

Among the 17 empirical papers in our final sample, we found that no research was conducted by or in collaboration with Indigenous peoples. Without study led by or conducted in collaboration with Indigenous people or groups, we cannot truly answer the question of whether or not solastalgia is a useful and or appropriate conceptual framework for understanding the lived experiences of climatic and environmental change among Indigenous peoples.

### 3.6. Strengths and Limitations 

To the best of our knowledge, this is the first study to map the published literature on solastalgia using a systematic process. A key strength of this work is the use of a systematic, transparent, and repeatable process. Also noteworthy, two independent reviewers assessed article eligibility (with very high agreement between reviewers) and data were extracted by one reviewer and verified by a second to minimize bias and enhance the robustness of our results [54]. Working with an experienced health sciences librarian and meaningful consultation with a diverse group of scholars, practitioners, and potential knowledge users at several stages of the scoping review process also enhanced the rigor of this study. There are however, several limitations that must also be acknowledged when interpreting the results reported here and when considering the implications of this work. First, our study did not search the grey literature, such that some characteristics of the emerging work on solastalgia—and practice- and policy-focused dimensions, specifically—may not be fully captured in this review. Additionally, not included in this review are solastalgia-related works of fiction, arts, popular culture, or the media. Second, we focused on English language publications only, which may have led to the exclusion of relevant papers published in other languages. A Spanish speaking co-author did search the Spanish literature and determined that there were no papers published in Spanish missed by our search strategy (as of October 2018). Third, we did not appraise the quality of individual papers included in our review. Although this may have allowed low-quality papers to be included in our review, this approach is consistent with guidance on scoping review methods [18,55]. Unlike systematic reviews, scoping reviews do not aim to assess the quality of papers reviewed [55]. Finally, we did not carefully consider the points of convergence and divergence between solastalgia and the range of other related concepts and areas of inquiry that also explore the lived experiences of loss, distress, and health consequences due to environmental change such as eco-anxiety, ecological grief, historical trauma, and mental and emotional impacts of climate change. This effort to focus may have led to the exclusion of a number of closely related papers; for example, ongoing work has identified sources such as Cunsolo et al.’s methodologically rigorous study combining EDS with qualitative methods on climate change among an Indigenous community [56].

## 4. Discussion

As noted previously, this review aimed to characterize and synthesize the scholarly literature on solastalgia. The data and key findings presented highlight important gaps in the literature and identify opportunities for theoretical, methodological, and practical advances concerning solastalgia. The following section discusses reflections on place that have emerged from the review process that merit further discussion. Subsequently, we present priority areas for future inquiry concerning solastalgia.

### 4.1. Reflections on Place and People–Place Relationships in the Solastalgia Literature

Place is a defining element of solastalgia, and people–place relationships are central to the ongoing study of the links between environmental change and human health and wellness. However, the ways in place and people–place relationships are understood are often unclear in the solastalgia literature. It is noteworthy that conceptual inconsistencies in terms of place-related concepts is common in other areas of inquiry as well [57]. Cunsolo and Ellis [33] have called for caution and clarity when applying place-related concepts in their review of ecological grief. We echo these calls and argue that ongoing study of solastalgia will benefit from greater clarity regarding how place and people–place concepts (i.e., place attachment, sense of place, connectedness to nature, etc.) are understood and applied.

The landscape construct was used by some papers in this review (e.g., [30,38,58]) and is increasingly used in the broader literature exploring the environment–health–place nexus as well [59]. According to Wylie, [46] landscape describes a process of place; “what we witness when we examine landscape is a process of continual interaction in which nature and culture both shape and are shaped by each other”. Landscapes include people, nature, and other beings, a perspective which aligns with the way in which many Indigenous scholars describe the reciprocal relationships between land, health, language, etc. [60,61]. Landscapes are not neutral backdrops where human activities unfold [46], rather they are relational, dynamic, and nested social-ecological systems. Grounding solastalgia research in landscapes allows for an emphasis on the richly intertwined aspect of nature and culture defining the places that people cherish while acknowledging the interplay across local, regional, and global scales, which we see as particularly relevant when aiming to understand the mental, emotional, and spiritual impacts of climate change—a global phenomenon that is witnessed and experienced in regional and local landscapes [22]. Thinking about places as landscapes may also help us to consider how individual and collective identity is wrapped up in place. Stobbelaar and Pedroli [62], define landscape identity as “the unique psycho-sociological perception of a place defined in a spatial–cultural space”, reminding us of both collective and individual aspects of landscape. Thus, landscape identity is the ability to see oneself in the on-going creation of landscapes and to acknowledge how one has been shaped by landscapes. A recent application of the idea of landscape identity in relation to environmental change is a study by Butler et al. [63] that considered landscape identity within an area altered by a catastrophic forest fire.

Drawing on the construct of landscape as described above, we define solastalgia as the distress caused by the unwelcome transformation of cherished landscapes resulting in cumulative mental, emotional, and spiritual health impacts. While initial solastalgia scales used a place descriptor of a specific location, in much of the solastalgia literature, place is described using the idea of ‘home’; we use ‘cherished landscapes’ instead. ‘Cherished’, to emphasize the deep emotional attachment to places that is common among those experiencing solastalgia. ‘Landscape’ to underline that places are dynamic, that humans are part of the biophysical world [64], and that place can be understood across nested scales and in multiple ways (e.g., place as home, place as specific location, place as local, place as regional, place as territory, place as global). Moreover, this shift from home to landscape in our understanding of solastalgia accounts for the possibility that individuals can experience solastalgia due to the undesired transformation of valued places other than their home environment. It is important to note that the specific way of defining and describing solastalgia presented here is grounded in, and particularly relevant to, the landscapes in which members of the International Solastalgia Collaborative live, learn, work, play, and study: the Lake Superior watershed. While we expect that this shift in the definition of solastalgia may be relevant to others engaging in solastalgia work across different landscapes, we acknowledge that it may not be relevant to all. 

### 4.2. Priority Areas for Future Inquiry

The research priorities presented below are not exhaustive, nor are they prescriptive. Instead, we offer research directions that have emerged from the scoping review process, highlighting what we see as most relevant to advancing solastalgia research overall as well as the application of solastalgia research specifically.

#### 4.2.1. Further Conceptual Clarity and Theoretical Development 

Solastalgia was developed “to give greater meaning and clarity to environmentally induced distress” [2]. The past 15 years of research have certainly contributed to growing conceptual clarity and theoretical advancement, as well as a more nuanced understanding of the emotional, mental, and spiritual health toll posed by our rapidly changing world. Nonetheless, additional research is needed to advance conceptual clarity and the theoretical foundations of solastalgia further. Efforts directed at clarifying the distinguishing features of solastalgia and unpacking the points of convergence and divergence between solastalgia and related concepts such as ecological grief, eco-anxiety, nature-deficit disorder, and mental health impacts of climate change, are particularly important. In their review of ecological grief, Cunsolo and Ellis [33] have similarly called for additional work to understand how ecological grief relates to solastalgia more fully. Albrecht [31,39,65] has done recent work towards developing nuances of solastalgia, grief, morning, and related emotional states including a “psychoterratic typology” and describing what he calls negative and positive Earth emotions [39]; this is a strong foundation to build on to advance conceptual clarity and theoretical development.

#### 4.2.2. Attention to Diversity and Gender Dimensions 

Another opportunity for advancing solastalgia study is to conduct research in a greater diversity of places and with a greater diversity of people and communities. This review highlighted that some regions of the world (e.g., global south) and specific population groups (e.g., youth, Indigenous peoples) are under-represented in the literature. Moreover, greater attention to the gendered dimensions of solastalgic experiences is warranted as the impacts of climatic and environmental change are “not gender neutral” [66]. Although a number of the papers included gender as a demographic variable, only three focused on the gendered dimensions of solastalgia in a substantive way. A related unpublished study sampled a random population of 1000 Lower Hunter New South Wales older residents (average age 66, range 55–86, equal men and women); researchers found an almost identical mean solastalgia score for men and women. Age in this truncated cohort study was not associated with solastalgia, but those with university degrees had statistically higher solastalgia scores compared with both those with technical college education and those with primary/secondary level educations [67]. It is clear from this review that gender is an understudied aspect of solastalgia research and an opportunity for advancement. 

Intersectionality [68,69] and recent work applying the lens of intersectionality to climate change research and action (e.g., Kaijser [70], Vinyeta et al. [71], Djouti et al. [72]) offers guidance for making progress towards paying attention to, and understanding, the ways in which gender, race, class and other identities intersect with experiences of solastalgia. Intersectionality “recognizes complex, horizontal (inter-community) and vertical (national, regional, local) interactions” while acknowledging context-specific mechanisms of exclusion and marginalization [72].

#### 4.2.3. Youth Voices 

A growing segment of the youth population is experiencing the impacts of climatic and environmental change in their everyday lives. Research has shown that young people may experience emotional, mental, and spiritual health impacts due to climate change [73,74,75]. Youth voices are an emerging source of power and inspiration in the face of climate change; the 2019 strikes for climate action initiated by high school students all across the planet is one concrete example of this [76]. Yet, despite youth concerns and responses, only one article in our review [77] examined youth-specific solastalgic experiences. Research using solastalgia as a theoretical framework with an explicit focus on youth is an opportunity to enhance our understanding of solastalgia as well as the overall study of place, environmental change, and human health. Participatory action research, which could promote youth empowerment and could set the stage for coping with solastalgia, is perhaps the most appropriate methodology for study examining youth experiences of solastalgia. The method of photovoice may be particularly useful. 

#### 4.2.4. Mediating Factors 

This scoping review has revealed that our collective understanding of the factors mediating the relationship between the lived experience of environmental change and solastalgia—such as place attachment, connectedness to nature, and sense of powerlessness—is limited. Although place attachment is very commonly hypothesized as a mediating factor; specifically that “greater feelings of place attachment results in more intense impact” [3], there is limited empirical evidence in support of this notion. The potential role of uncertainty as a mediating factor, more specifically that uncertainty about the future due to climatic and environmental change, is also under-examined in the literature and is particularly relevant concerning the impacts of climate change. 

A subset of articles from this review considered the relationship between environmental change, powerlessness, and solastalgia [2,3,32,44]. Arguably, a perceived or actual inability to prevent the transformation of cherished landscapes, and therefore a sense of powerlessness, may exacerbate solastalgia. Askland and Bunn [48], whose paper aims to “critically engage with the topic of solastalgia including a consideration of power and dispossession” argue that “whilst solastalgia may help explore the relationship between environmental and human distress, the sense of displacement and loss that emerge are entangled with questions of power and dispossession beyond the biophysical realm". Along a similar vein, Albrecht et al. [2] write about the lived experiences of persistent drought as well as experiences of large-scale open-cut coal mining in Australia: “In both cases, people exposed to environmental change experienced negative affect that is exacerbated by a sense of powerlessness or lack of control over the unfolding process.” Also noteworthy, we found that the terms “unwanted”, “imposed”, and “unwelcome” are commonly employed when describing a source of environmental change causing solastalgia. These descriptive terms highlight the role that powerlessness (or perceived powerlessness) can play in one’s experience of environmental change and degradation of cherished landscapes. Nevertheless, the idea of power/powerlessness as a mediating factor of solastalgic experiences is under-investigated. How, for instance, does one’s positionality within society, access to resources, and decision-making power determine the likelihood or intensity of solastalgia? 

#### 4.2.5. Accounting for the Cumulative Nature of Health Impacts

Ellis and Albrecht [35] argue that solastalgic experiences often come “on top of other economic and social stressors”, thereby exacerbating health disadvantages and inequities [35]. We certainly agree with this notion and see evidence of this in our review. The research aimed at understanding how multiple factors interact with the transformation of cherished landscapes, across both time and space, resulting in cumulative impacts on emotional, mental, and spiritual health is a priority moving forward. Integrating the work on historical trauma [78,79] into future solastalgia research could be a useful avenue for advancing our understanding of the cumulative nature of impacts. Historical trauma, understood as the cumulative and intergenerational experiences of multiple chronic traumas experienced by Indigenous peoples (and others) including, but certainly not limited to, land dispossession and degradation [78,79]. Solastalgic distress and other mental, emotional, and spiritual health impacts caused by environmental change “can be understood in light of historical trauma losses and disruptions tied to place or land” [80]. The emerging literature on the cumulative environmental, community, and health impacts of resource development (e.g., Parkes [81], Gillingham [6]) could also be useful for scholars aiming to examine the cumulative nature of health impacts that emerge as a consequence of the transformation of cherished landscapes. 

#### 4.2.6. Additional Mixed-Methods and Quantitative Research 

Most of the empirical papers reviewed here used qualitative methods, specifically interviews, to collect data on the lived experiences of solastalgia. Qualitative methods are undoubtedly appropriate and useful, however, the solastalgia literature overall would benefit from additional quantitative and mixed-method research. A specific area for advancement is further development, validation, and application of scales to quantitatively measure solastalgia. The EDS developed by Higginbotham [3] and adapted by Warsini [42,43] and Eisenman [34] is a strong foundation to build on. Additional testing, across diverse landscapes and populations, and in response to a greater variety of sources of hazards (both acute and chronic), is warranted. Note, a number of masters and PhD studies have made use of EDS/solastalgia framework, and many of these have not shown up in the 29 studies (outside the scoping criteria); these additional sources may be able to provide support and/or critique to the use of the EDS [67]. Further, population-based studies using quantitative data collection and analysis methods could help enhance our limited understanding of mediating factors as described above, risk factors, and vulnerable sub-groups, while also helping to address an important and yet unanswered question: what is the burden of solastalgia at the population level?

#### 4.2.7. Research Led by or Conducted in Partnership with Indigenous Communities/Peoples 

In early and foundational discussions of solastalgia, scholars took care to describe the nature of Indigenous peoples’ and non-Indigenous peoples’ relationships to the land [1,2,30,32]. In his early work, Albrecht [1] described Indigenous experiences of distress related to environmental change as amplified given that “they (Indigenous peoples) live through the destruction of their cultural traditions and their lands”. In many cases, Indigenous people’s sense of personal and cultural identity derives from their relationships with the land (including water and non-human beings). Indigenous peoples suffer from distress when their ties to those places (or the places themselves) change, inferring that such distress may be experienced differently than the place-related distress of non-Indigenous people. However, there are other factors—in addition to strong relationships with the land—that are common among Indigenous populations, and that may impact Indigenous peoples’ experiences of solastalgia. Indigenous people’s collective and layered experiences of historical trauma like land dispossession, colonization and forced assimilation, boarding/residential schools, and broken nation-to-nation treaties may also shape experiences and responses to climatic and environmental change, or the intensity of experiences of solastalgia [49,82,83]. 

Importantly, solastalgia may be an inadequate or even inappropriate way of describing Indigenous peoples’ experiences of distress related to changes to their cherished landscapes. In our consultation process, an Indigenous scholar asserted that “solastalgia is a colonized word, and using the term solastalgia (to describe Indigenous experiences) feels like trying to knock a square peg into a round hole. There are Indigenous concepts that can be used to describe solastalgia-like experiences better” [84]. In some applications, solastalgia has been used to describe individual, rather than collective, experiences of distress, emphasizing an “individualism and colonial isolation that settler colonialism fosters” [85]. It is also important to acknowledge that there is an intimate and reciprocal relationship between Indigenous languages and the landscapes and places in which they develop: “Indigenous Knowledge comes from the land through the relationships Indigenous People develop and foster with the essential forces of nature. These relationships are encoded in the structure of Indigenous languages and in Indigenous political and spiritual systems. They are practiced in traditional forms of governance, and they are lived in the hearts and minds of Indigenous People” [49]. Therefore, the concept of solastalgia may be best articulated in the specific language(s) of the peoples who are experiencing it. Albrecht [30] provides some examples of this, including the Baffin Island Inuit term *uggianaqtuq*, which has connotations of “a friend acting ‘strangely’ or in an unpredictable way”. There are other bodies of literature that use different conceptual frameworks and theories to understand and describe Indigenous peoples’ experiences of distress related to environmental change and loss, including historical trauma, ecological grief, and the mental health impacts of climate change (sources). The work of Cunsolo et al. [13,86] in Canada’s far North is an excellent example of community-led research examining the mental health implications of climatic change, in partnership with Inuit peoples. Another example is the research conducted by Rigby et al. [87], which explored the impact of prolonged drought on the emotional well-being of Aboriginal communities in rural Australia, in collaboration with Aboriginal partners. 

Some of these critical considerations could be addressed through research led-by, or conducted in partnership with, Indigenous scholars and Indigenous communities. Non-Indigenous scholars and practitioners seeking to understand Indigenous peoples’ experiences of solastalgia may benefit from using “two-eyed seeing” [88] and decolonizing methodologies to inform and guide their collaborative solastalgia research. “Two-eyed seeing” is a framework that embraces both Indigenous and western ways of knowing in balance with one another and has the power to “reshape the nature of the questions we ask in the realm of Indigenous health research” [88]. Decolonizing methodologies encourage scholars to care for the land and stewards of the land while also encouraging critical questions throughout research processes to “enhance, rather than simply describe or define, the lives of Indigenous peoples” [89,90]. Ultimately, it is for Indigenous researchers and their communities to decide if the idea of solastalgia is useful to express experiences of negative environmental change or if existing concepts from Indigenous scholarship are more appropriate. 

#### 4.2.8. Practical Implications and Applications

Our scoping review process has underscored that the practical implications and the potential applications of solastalgia research are generally overlooked at present. Potential practical applications of solastalgia include its use in health and environmental impact assessment processes [51], clinical settings (i.e., social work, psychology), and climate change and health vulnerability assessments. How to integrate solastalgia into these practical arenas remains unclear and doing so will require further theoretical development in addition to the development of validated scales to empirically measure and monitor solastalgic experiences. 

There is a rich scholarly history documenting that connectedness to nature is critical to health and wellness [91,92,93,94]. Therefore, an important practical implication of a better understanding of solastalgia is greater recognition of the importance of safeguarding access to nature. Efforts to expand equitable access to nature must be taken as part of a response to the distress experienced by negative transformations of cherished landscapes. Our interconnectedness with nature may not only be at the core of our distress, but also key to coping with unwelcome transformations we experience. Given the centrality of place in the lived experiences of solastalgia, interventions aimed at responding to, and coping with, solastalgic distress will have to be informed by an in-depth understanding of people–place relationships across unique landscapes. 

Key questions also remain in terms of how to address solastalgia. For instance, how can individuals and groups cope with solastalgic distress? Can the restoration of degraded landscapes help people cope with distress and enable healing? Can interventions focused on collective healing enable coping with solastalgia in ways that promote human and ecosystem health? These questions will be a central focus of the International Solastalgia Collaborative in the Lake Superior basin moving forward. We believe that responses to solastalgia have the potential to simultaneously heal human and ecosystem health while inspiring collective action against the structural forces underpinning climatic and environmental change; how exactly we can and should go about this in practice remains untested and under-investigated. We also believe that benefits can be derived from exploring the emotional dimensions of climatic and environmental change more fully. Emotions are defined broadly as “that which moves us” [39]; it is often affect, rather than knowledge or information alone, that moves people to take action in the face of climatic and environmental change [95,96]. 

## 5. Conclusions

Solastalgia is an increasingly useful concept for understanding the links between ecosystem health and human health, specifically, the cumulative impacts of climatic and environmental change on mental, emotional, and spiritual health. Given the speed and scale of climate change and the unbridled advancement of resource extraction, more and more people will experience the unwelcome transformation of cherished landscapes and solastalgic distress. Consequently, we anticipate that solastalgia will be increasingly applied, developed, and measured across a range of scholarly arenas. 

This study reviewed 15 years of scholarly literature on solastalgia using a scoping review process. Drawing on our findings, we have offered several priorities to advance the study of solastalgia and its application. Overall, we identify a need for additional research using diverse methodologies and data collection methods, across a greater diversity of people and landscapes, and conducted in collaboration with affected populations and potential knowledge users, alongside greater attention to the practical implications and applications of solastalgia research. Further conceptual clarity and theoretical development is called for. More work is needed to advance our collective understanding of mediating factors and to identify the points of convergence and divergence between solastalgia and related areas of inquiry such as ecological grief and eco-anxiety in particular. Two key outcomes of this study include our use of the landscape construct in relation to solastalgia and a call to better understand Indigenous peoples’ lived experiences of landscape transformation and degradation in the context of historical traumas. Finally, whether solastalgia is a useful concept for understanding the mental, emotional, and spiritual health impacts of environmental and climatic change from the perspective of Indigenous peoples themselves remains unclear. 

As Albrecht himself reminds us, solastalgia is “a work in process” as the “novel effects of grand-scale environmental damage to places, hearts and psyches become evident” [39]. Our review has highlighted that there is still much to be learned about solastalgia.

## Figures and Tables

**Figure 1 ijerph-16-02662-f001:**
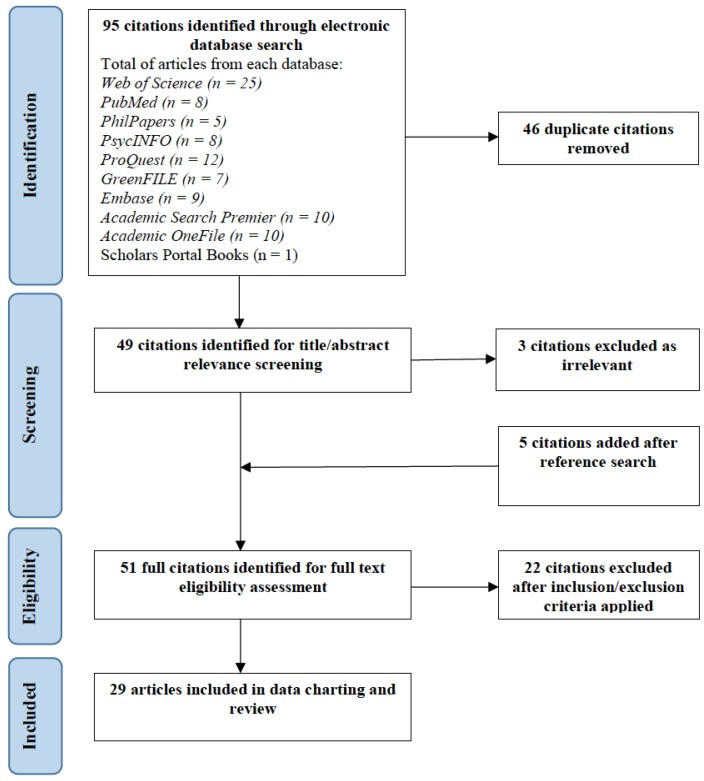
Adapted PRISMA diagram. PRISMA: Preferred Reporting Items for Systematic Reviews and Meta-Analyses.

**Figure 2 ijerph-16-02662-f002:**
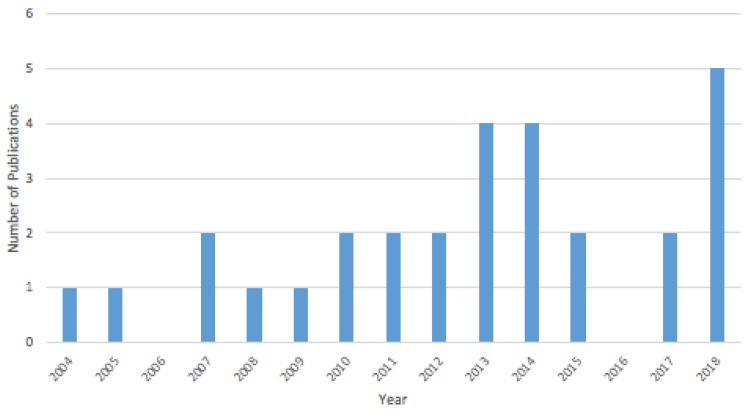
Year of publication, 2004 to October 2018.

**Table 1 ijerph-16-02662-t001:** Inclusion and exclusion criteria.

Inclusion Criteria	Exclusion Criteria
Dates: 2003 to 2018 (2003 was chosen as the earliest date because the term was first introduced by an Australian environmental philosopher to an Ecohealth Conference in Montreal in 2003)	Language (non-English articles excluded) Periodicals, news items, op-eds No explicit focus on place No substantive discussion of, or reference to, health/well-being/wellness No explicit empirical, theoretical, or practical focus in relation to solastalgia

**Table 2 ijerph-16-02662-t002:** Sources of environmental change causing solastalgia ^1.^

Sources of Environmental Change Causing Solastalgia	N
Extreme weather event/natural disaster (e.g., floods, droughts, hurricane)	16
Climate change	13
Prolonged environmental transformation	13
Land clearing/deforestation	12
Resource extraction/development (e.g., mining)	9
Gentrification and/or changing built environment	6
Displacement or appropriation of land/political violence/war	6
Rapid industrial development	4

^1^ Papers could be coded for more than one factor, so the total is greater than 29.

**Table 3 ijerph-16-02662-t003:** EDS items operationalizing solastalgia [3].

Feelings of Solastalgia from Environmental Change
Sad when looking at degraded landscapes and mine voids
Farming lifestyle depending on good land and water is threatened by change
Worried that valued aspects of place—clean air and water, scenery—are being lost
Unique aspects of nature in this place are being lost
Miss peace and quiet once enjoyed in this place
Sad that familiar animals and plants are disappearing
Ashamed of the way this area looks now
Thought of my families being forced to leave this place upsets me
Sense of belonging undermined by change

Note: EDS: Environmental Distress Scale.

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
