# Peer review of "Mapping the Solastalgia Literature: A Scoping Review Study"

_ijerph, 2019, doi:10.3390/ijerph16152662_

Round 1
Reviewer 1 Report
Though I'm not familiar with "scoping reviews", the methodology here seems well thought-through, and appropriate to the research questions the authors have set for themselves. It's not adopted uncritically from the existing literature, moreover; the authors show that they've carefully thought about what best suits their own enquiry and tailor their method accordingly based on several resources.
I felt that the results might be more clearly presented in section 3. This is sub-divided into the various questions that the authors posed, which is appropriate. Nevertheless, one has to read the entirety of each section to get a sense of what the answer to each question is. Obviously, readers should do this; but many won't. An "exectuve summary" or abstract of the findings would help, either at the start of section 3 as a whole covering all research questions, or in a sentence or two at the start or end of each sub-section. The sub-section on "strengths and limitations" is appropriately reflective.
The "discussion" section (section 4) seems thoughtful and appropriate to me; it will hopefully map out a course for future studies of what - the authors successfully make the case in their introduction - is an urgent and important phenomenon.
Author Response
ijerph-553597
Response to reviewers
We would just like to thank the three reviewers for thoughtful, supportive, and encouraging review. I hope the following revisions and responses will satisfy concerns and help us present a stronger paper. Responses in blue.
Reviewer 1
Though I'm not familiar with "scoping reviews", the methodology here seems well thoughtthrough, and appropriate to the research questions the authors have set for themselves. It's not adopted uncritically from the existing literature, moreover; the authors show that they've carefully thought about what best suits their own enquiry and tailor their method accordingly based on several resources.
Thank you, appreciated.
I felt that the results might be more clearly presented in section 3. This is sub-divided into the various questions that the authors posed, which is appropriate. Nevertheless, one has to read the entirety of each section to get a sense of what the answer to each question is. Obviously, readers should do this; but many won't. An "exectuve summary" or abstract of the findings would help, either at the start of section 3 as a whole covering all research questions, or in a sentence or two at the start or end of each sub-section.
The answers to the questions we ask are not straight-forward, but rather complicated and detailed. While I think the suggestion is a good one, however, we are trying to balance overall length with providing the details of our analysis. A few sentences have been added, and other already existing sentences from the text are noted here to provide the Reviewer with evidence of our attempt to satisfy this concern:
3.1
Note changes. Line 167: Key findings from exploration of the general characteristics of the Solastalgia literature show the literature to be recent, multidisciplinary and empirical in nature, and further, the Australian origins of the work is clear.
3.2
No change.
3.4:
Line 274: (existing sentence, no change) Nearly all papers in this review (27/29) specifically describe solastalgia as a place-based phenomenon. Interestingly, despite the centrality of place in relation to solastalgia, few authors clearly and explicitly define what they mean by “place”.
3.5:
Line 309: (existing sentence, no change) Of the 29 papers reviewed, nearly all (approximately 90%) examined solastalgia either through a western worldview or through multiple worldviews, rather than using an Indigenous worldview specifically (or multiple Indigenous worldviews).
The sub-section on "strengths and limitations" is appropriately reflective.
The "discussion" section (section 4) seems thoughtful and appropriate to me; it will hopefully map out a course for future studies of what - the authors successfully make the case in their introduction - is an urgent and important phenomenon.
Thank you—exactly our intentions.
Reviewer 2 Report
OVERALL COMMENTS
I was very pleased to see an in-depth examination of the solastalgia literature, and agree with the authors that now is indeed the time for an in-depth consideration of the concept. This is a strong manuscript, with some interesting findings and synthesis. I think it will be a useful resource for people in diverse fields and disciplines, as well as policy and practice.
I have minor comments/edits, but believe this paper should be published, and will make a positive contribution to the literature.
ABSTRACT
The abstract mentions climate change and resource extraction – which are of course extremely important. But don’t forget loss of biodiversity, pollution, deforestation, desertification, etc. – all of these shifts lead to solastalgia. It’s important to not limit to just climate change and extractive activities. Further down in your article you do indicate that the literature ties to other topics, such as flooding, drought, etc., as well as things like displacement and gentrification. It would be good to link this more thoroughly in your abstract.
METHODS
The methods are clear and well-written, and do a good job of following the Scoping Review standards.
Why did your search criteria require a specific focus on place and on health/wellbeing? Please explain further in the Methods why that was a specific requirement, as it is not fully clear in the current description, or in the Introduction leading up to the Methods.
In particular, the four research questions do not emphasize health or wellbeing in the foci, which then doesn’t align with removing articles that don’t focus on health and wellbeing. It’s important to make sure your overall questions match the categories in the inclusion/exclusion criteria.
Figure 1 is actually a result, and should be moved to the Results section.
RESULTS
Insert Figure 1 in the Results.
Page 6, line 210: please insert the references that use those words to describe solastalgia.
Author Response
ijerph-553597
Response to reviewers
We would just like to thank the three reviewers for thoughtful, supportive, and encouraging review. I hope the following revisions and responses will satisfy concerns and help us present a stronger paper. Responses in blue.
Reviewer 2
OVERALL COMMENTS
I was very pleased to see an in-depth examination of the solastalgia literature, and agree with the authors that now is indeed the time for an in-depth consideration of the concept. This is a strong manuscript, with some interesting findings and synthesis. I think it will be a useful resource for people in diverse fields and disciplines, as well as policy and practice. I have minor comments/edits, but believe this paper should be published, and will make a positive contribution to the literature.
Thank you for your support of our efforts.
ABSTRACT
The abstract mentions climate change and resource extraction – which are of course extremely important. But don’t forget loss of biodiversity, pollution, deforestation, desertification, etc. – all of these shifts lead to solastalgia. It’s important to not limit to just climate change and extractive activities. Further down in your article you do indicate that the literature ties to other topics, such as flooding, drought, etc., as well as things like displacement and gentrification. It would be good to link this more thoroughly in your abstract.
Note changes. Line 12: Given the speed and scale of climate change alongside biodiversity loss, pollution, deforestation, resource extraction, and other environment challenges, more and more people will experience solastalgia.
METHODS
The methods are clear and well-written, and do a good job of following the Scoping Review standards. Why did your search criteria require a specific focus on place and on health/wellbeing? Please explain further in the Methods why that was a specific requirement, as it is not fully clear in the current description, or in the Introduction leading up to the Methods. 3 In particular, the four research questions do not emphasize health or wellbeing in the foci, which then doesn’t align with removing articles that don’t focus on health and wellbeing. It’s important to make sure your overall questions match the categories in the inclusion/exclusion criteria.
It is our opinion that we have made a case for the environment-health-place nexus, for example we include text such as the following in the introduction, Line 30: Solastalgia, preliminarily and broadly defined as the distress caused by the transformation and degradation of one’s home environment [1–3] is a relatively new concept with particular relevance to the environment-health-place nexus. Given the current speed and scale of climatic and environmental change, and the increasingly urgent calls to better understand the links between changing environments and human health, solastalgia merits further investigation and theoretical development [4–6]. As Louv [7] argues, “[i]f climate change occurs at the rate that some scientists believe it will, and if human beings continue to crowd into de-natured cities, then solastalgia will contribute to a quickening spiral of mental illness” [7].
-and-
Line 50: At the core of this mounting evidence, the growing media attention, and the increasingly common lived experiences of climatic and environmental change, comes growing distress and a range of mental, emotional, and spiritual health consequences.
Figure 1 is actually a result, and should be moved to the Results section.
Note changes. Figure 1 has been moved to line 163.
RESULTS
Insert Figure 1 in the Results.
Note changes. Figure 1 has been moved to Line 163
Page 6, line 210: please insert the references that use those words to describe solastalgia
Note changes. Line 215: Sources have been added.
Reviewer 3 Report
239-241 If the authors have access to the actual survey questions, they might wish to consider concisely presenting the common/core solastalgia scale items that were used. These are the ‘operational definitions’ of the concept that researchers employed to map its dimensions.
248: Important to cite Cunsolo et al (2012), early methodologically rigorous study combining EDS with qualitative methods on climate change among an Indigenous community.
Cunsolo, A.W., Harper, S.L., Ford, J.D., et al., “From this place and of this place:” Climate change, sense of place, and health in Nunatsiavut, Canada. Social Science and Medicine, 2012 (75(3), 538-547
396 Expanding on my comments for 239-241, what form of words were used to encapsulate the specific environment that the respondents were asked to think about? Was it ‘your home’, the ‘local environment where you live’, ‘your neighbourhood’, ‘local area’, or a specific wetland, etc? This definition gets to the question of scale, specificity and landscape, that is discussed here. Such information may not be found within the publications, however.
403 Authors speak of a shift from ‘home’, to ‘landscape’. Initial Solastalgia scales themselves did not begin with a restriction on ‘home,’ but rather a specific location, i.e., “Change in your local environment in the Singleton area.”
435-439. Ethnographic studies on gender and solastaligia (and other diversity aspects) are essential.
An unpublished randomly sampled population study recruiting 1000 Lower Hunter New South Wales older residents (average age 66, range 55-86; equal men and women) found an almost identical mean solastalgia score for men and women. Age in this truncated cohort study was not associated with solastlagia, but those with university degrees had statistically higher solastalgia scores compared with both those with technical college education and those with primary/secondary level educations. I can make these data available to the authors if they wish to note them as a Personal Communication.
462. Sense of powerlessness and environmental injustice was a dominant theme in the Upper Hunter mine affected interviews, not surprising given the obliteration of landscape by some of the world’s largest corporations backed up by a state government hungry for mining royalties. Expression of Solastalgia and powerlessness were inseparably intertwined.
This dimension was added to later versions of the solastalgia scale and attained high homogeneity with the other items.
477. the word ‘effect’ should really be ‘affect’ and is possibly misspelled in the original paper.
510. Population based quantitative studies. The EDS needs adaptation to each site/cultural context, and never was an off the shelf instrument, and this applies to the solastalgia sub-scale.
A number of masters and PhD studies have made use of EDS/solastalgia framework, and many of these have not shown up in the 29 studies, no doubt because they were not published.
It is accepted that research theses were outside the scoping criteria. However, a survey of research theses that use and further refine the concept of solastalgia, could be most useful. The uptake in new research theses would be a good indicator of the perceived use/value of this concept in academia. For example, the two papers most often cited on solastalgia now have 716 citations between them, many of them research theses.
426. Albrecht published a 2017 book chapter for an anthology that captured more of the nuances of solastalgia, grief, mourning and related emotional states: Albrecht, Glenn. A. (2017) “Solastalgia and the New Mourning.” In Mourning Nature: Hope at the Heart of Ecological Loss & Grief, edited by Ashlee Cunsolo and Karen Landman, 292-215. Montreal: McGill-Queen's University Press. It would be good if the authors could cite this publication on solastalgia, grief and mourning here in this section on conceptual clarity.
529-531. The cross-cultural validity and universality of solastalgia were never assumed in its first conception and measurement. Albrecht always characterised solastalgia as a new concept in the English language. It is for Indigenous researchers and their communities to decide if this notion is useful to express their experiences of negative environmental change. Learning Indigenous language expressions in this space is an invaluable and high priority development for this field.
534. The author’s cite Simpson to state that Solastalgia has generally been used to describe individual, rather than collective, experiences of distress, emphasizing an “individualism and colonial isolation that settler colonialism fosters”.
This characterisation does not seem justified if it is directed toward Albrecht’s own writing and activism. He and colleagues have always worked within communities. He gave expert testimony before the NSW Land and Environment Court, about solastalgia. He did so on behalf of a community action group fighting a massive coal mine expansion, in order to establish the environmental distress of the village residents. Moreover, much of the solastalgia literature actually reflects "community-based" studies.
559 Practical applications.
A recent workshop at Sydney University bringing social impact assessors together with state environmental planners concluded with a recommendation to include environmental distress measurement in planning decisions. The challenge is to bring forward systematic methods for evaluating planning proposals and the body of knowledge about environmental distress and solastalgia in the literature. This article will help this cause.
Solastalgia, the arts and popular culture. While excluded from the aims of this paper, there is now a wealth of information demonstrating how solastalgia has been taken up in the art world and popular culture generally. Missy Higgins, one of Australia’s most popular singers, named her 2018 album, Solastalgia, and captures many of the definitions used in this scoping paper:
https://www.abc.net.au/doublej/featured-music/feature-albums1/missy-higgins-solastalgia/10265986
Author Response
ijerph-553597
Response to reviewers
We would just like to thank the three reviewers for thoughtful, supportive, and encouraging review. I hope the following revisions and responses will satisfy concerns and help us present a stronger paper. Responses in blue.
Reviewer 3:
239-241 If the authors have access to the actual survey questions, they might wish to consider concisely presenting the common/core solastalgia scale items that were used. These are the ‘operational definitions’ of the concept that researchers employed to map its dimensions.
Note changes. Line 255: Items added to enrich the presentation of the EDS, see Table 2.
248: Important to cite Cunsolo et al (2012), early methodologically rigorous study combining EDS with qualitative methods on climate change among an Indigenous community. Cunsolo, A.W., Harper, S.L., Ford, J.D., et al., “From this place and of this place:” Climate change, sense of place, and health in Nunatsiavut, Canada. Social Science and Medicine, 2012 (75(3), 538-547
Note changes. Line 364: This article is now referenced in the limitations section as an example of articles we may have missed due to our criteria:
This effort to focus may have led to the exclusion of a number of closely related papers, for example on going work has identified sources such as Cunsolo et al.’s methodologically rigorous study combining EDS with qualitative methods on climate change among an Indigenous community.
396 Expanding on my comments for 239-241, what form of words were used to encapsulate the specific environment that the respondents were asked to think about? Was it ‘your home’, the ‘local environment where you live’, ‘your neighbourhood’, ‘local area’, or a specific wetland, etc? This definition gets to the question of scale, specificity and landscape, that is discussed here. Such information may not be found within the publications, however.
We did not find this information. We would have to get access to interview guides and notes from the research not included in papers.
403 Authors speak of a shift from ‘home’, to ‘landscape’. Initial Solastalgia scales themselves did not begin with a restriction on ‘home,’ but rather a specific location, i.e., “Change in your local environment in the Singleton area.”
Note changes. Line 407: While initial solastalgia scales used a place descriptor of a specific location, in much of the solastalgia literature, place is described using the idea of ‘home’; we use ‘cherished landscapes’ instead.
Line 412: place as specific location
435-439. Ethnographic studies on gender and solastaligia (and other diversity aspects) are essential. An unpublished randomly sampled population study recruiting 1000 Lower Hunter New South Wales older residents (average age 66, range 55-86; equal men and women) found an almost identical mean solastalgia score for men and women. Age in this truncated cohort study was not associated with solastlagia, but those with university degrees had statistically higher solastalgia scores compared with both those with technical college education and those with primary/secondary level educations. I can make these data available to the authors if they wish to note them as a Personal Communication.
Yes, thank you. We would like to use this information as personal communication. Can your identity be revealed for our references? We have left reference #66 as a place holder for the proper name.
Note changes. Line 449: A related unpublished study sampled a random population of 1000 Lower Hunter New South Wales older residents (average age 66, range 55-86, equal men and women); researchers found an almost identical mean solastalgia score for men and women. Age in this truncated cohort study was not associated with solastalgia, but those with university degrees had statistically higher solastalgia scores compared with both those with technical college education and those with primary/secondary level educations.
462. Sense of powerlessness and environmental injustice was a dominant theme in the Upper Hunter mine affected interviews, not surprising given the obliteration of landscape by some of the world’s largest corporations backed up by a state government hungry for mining royalties. Expression of Solastalgia and powerlessness were inseparably intertwined. This dimension was added to later versions of the solastalgia scale and attained high homogeneity with the other items.
477. the word ‘effect’ should really be ‘affect’ and is possibly misspelled in the original paper.
Note changes. Line 495: Effect changed to affect. I checked the original and it was correct; the mistake was ours.
510. Population based quantitative studies. The EDS needs adaptation to each site/cultural context, and never was an off the shelf instrument, and this applies to the solastalgia sub-scale. A number of masters and PhD studies have made use of EDS/solastalgia framework, and many of these have not shown up in the 29 studies, no doubt because they were not published. It is accepted that research theses were outside the scoping criteria. However, a survey of research theses that use and further refine the concept of solastalgia, could be most useful. The uptake in new research theses would be a good indicator of the perceived use/value of this concept in academia. For example, the two papers most often cited on solastalgia now have 716 citations between them, many of them research theses.
Note changes. Line 528: Note, a number of masters and PhD studies have made use of EDS/solastalgia framework, and many of these have not shown up in the 29 studies (outside the scoping criteria); these additional sources may be able to provide support and/or critique to the use of the EDS.
May we use a personal communication for this information as well? (Again, #66).
426. Albrecht published a 2017 book chapter for an anthology that captured more of the nuances of solastalgia, grief, mourning and related emotional states: Albrecht, Glenn. A. (2017) “Solastalgia and the New Mourning.” In Mourning Nature: Hope at the Heart of Ecological Loss & Grief, edited by Ashlee Cunsolo and Karen Landman, 292-215. Montreal: McGill-Queen's University Press. It would be good if the authors could cite this publication on solastalgia, grief and mourning here in this section on conceptual clarity.
Note changes. Line 436: Albrecht [30,33,64] has done recent work towards developing nuances of solastalgia, grief, mourning and related emotional states including a…
(source added: Albrecht, Glenn. A. (2017) “Solastalgia and the New Mourning.” In Mourning Nature: Hope at the Heart of Ecological Loss & Grief, edited by Ashlee Cunsolo and Karen Landman, 292-215. Montreal: McGill-Queen's University Press.)
529-531. The cross-cultural validity and universality of solastalgia were never assumed in its first conception and measurement. Albrecht always characterised solastalgia as a new concept in the English language. It is for Indigenous researchers and their communities to decide if this notion is useful to express their experiences of negative environmental change. Learning Indigenous language expressions in this space is an invaluable and high priority development for this field.
Note changes. Line 579: Ultimately, It is for Indigenous researchers and their communities to decide if this notion is useful to express their experiences of negative environmental change or if existing concepts from indigenous scholarship are more appropriate.
534. The author’s cite Simpson to state that Solastalgia has generally been used to describe individual, rather than collective, experiences of distress, emphasizing an “individualism and colonial isolation that settler colonialism fosters”. This characterisation does not seem justified if it is directed toward Albrecht’s own writing and activism. He and colleagues have always worked within communities. He gave expert testimony before the NSW Land and Environment Court, about solastalgia. He did so on behalf of a community action group fighting a massive coal mine expansion, in order to establish the environmental distress of the village residents. Moreover, much of the solastalgia literature actually reflects "community based" studies.
Note changes. Line 555: In some applications, Solastalgia has also been used to describe individual, rather than collective, experiences of distress, emphasizing an “individualism and colonial isolation that settler colonialism fosters” [79]
559 Practical applications. A recent workshop at Sydney University bringing social impact assessors together with state environmental planners concluded with a recommendation to include environmental distress measurement in planning decisions. The challenge is to bring forward systematic methods for evaluating planning proposals and the body of knowledge about environmental distress and solastalgia in the literature. This article will help this cause.
It is so affirming to hear that this work will have such application, thank you for this information. Our intent was to help push the conversation forward.
Solastalgia, the arts and popular culture. While excluded from the aims of this paper, there is now a wealth of information demonstrating how solastalgia has been taken up in the art world and popular culture generally. Missy Higgins, one of Australia’s most popular singers, named her 2018 album, Solastalgia, and captures many of the definitions used in this scoping paper: https://www.abc.net.au/doublej/featured-music/feature-albums1/missy-higginssolastalgia/10265986
Thank you for this, I am listening to the album via Spotify as I work on these revisions. The song “How was I to know”… How was I to know that I would be a stepping stone to the end, to the end of everything?… brings tears to my eyes. I guess the sentiment in her final hopeful comment is what guides our work… turning the stepping stones around to everything that matters…